# Time-Impact Optimal Trajectory Planning for Wafer-Handling Robotic Arms Based on the Improved Snake Optimization Algorithm

**DOI:** 10.3390/s25061872

**Published:** 2025-03-18

**Authors:** Yujie Ji, Jiale Yu

**Affiliations:** School of Mechanical Engineering, Shenyang Ligong University, Shenyang 110159, China; yjl2801036037@163.com

**Keywords:** ISO algorithm, S-shaped speed curve, trajectory planning, time-impact optimal

## Abstract

To enhance the working efficiency of a wafer-handling robotic arm and simultaneously alleviate the impact and vibration during the motion process, a trajectory planning approach based on an improved snake optimization (ISO) algorithm is proposed. The following improvements have been made to the snake optimization (SO) algorithm: the introduction of a Chaotic Tent Map for initializing the swarm, the use of randomly perturbed dynamic learning factors to replace fixed values, the application of a cosine annealing learning rate for self-adaptively updating individual positions, and the incorporation of Bayesian optimization for parameterization and fine-tuning of the system’s selection process. Furthermore, the ISO algorithm is applied in the Cartesian space of the robotic arm to effectively address the trajectory planning challenge of the single-segment start–stop S-shaped speed curve with arc transitions. The simulation results indicate that the improved S-shaped speed curve has increased by 24.1% compared with the original plan, and the mean and variance rankings of ISO algorithm have, respectively, improved by 60.8% and 63.4%, compared with the SO algorithm. Meanwhile, this study has successfully achieved the Pareto optimal solution with time and impact as the targets based on the established MATLAB experimental simulation platform.

## 1. Introduction

With the rapid advancement of the semiconductor industry, the optimal trajectory planning problem for manipulator arms has become a critical research focus [1,2,3,4]. It is a fundamental and critical challenge in the field of robotic manipulators. At its core, the goal of optimal trajectory planning is to determine a motion path for the manipulator that meets specific requirements such as minimizing operation time, energy consumption, and mechanical stress, while ensuring the accuracy of reaching the target position and orientation. To achieve the overarching objective, the following three critical aspects should be thoroughly considered: the selection of an appropriate operational space for the robotic arm, the design of suitable trajectory curves, and the implementation of an efficient optimization algorithm.

When deliberating the trajectory planning of the manipulator arm, a choice must be made between the planning schemes in Cartesian space and Joint space, which are typically determined by specific task requirements, characteristics of the manipulator arm, and tracking accuracy, among other factors. For instance, trajectory planning in Cartesian space enables direct control of the position and orientation of the manipulator end effector, which is relatively more intuitive than that in Joint space, and can more effectively plan smooth motion trajectories [5,6,7]. From the perspective of advanced computational methods, Xiao et al. [8] successfully mitigated the shock and vibration experienced by a robot during its movement from the starting position to the target position in Cartesian space, thereby ensuring the accuracy and safety of point-to-point movements. In contrast, Zhang et al. [9] successfully planned two sets of trajectories using polynomial interpolation in Joint space and verified the model by using SimMechanics. However, owing to the insufficient robustness of the PID controller against external environmental factors, the simplification of the model significantly affected the accuracy of the experiment. Additionally, the improved quantum particle swarm optimization algorithm proposed by Luo et al. [10] performed outstandingly in the trajectory planning of the robot joint space, effectively enhancing the efficiency and accuracy of trajectory planning. Regarding the S-shaped speed curve, Wu et al. [11] investigated how to plan trajectories in Joint space of industrial robots that satisfy kinematic constraints and have continuous jerk. Nevertheless, trajectory planning in Joint space requires complex inverse kinematic solutions. Due to joint coupling effects, trajectories converted to Cartesian space may exhibit distortions or deviations from the desired path. This constitutes a potential issue for manipulator arms employed in wafer-handling scenarios.

Trajectory planning curves are vital for motion control strategies. The common types include linear interpolation, B-spline curves, polynomial curves, T-shaped speed curves, and S-shaped speed curves. Polynomial interpolation is frequently employed for this purpose. By selecting the appropriate polynomial order, its adaptability can be flexibly adjusted to match different numbers of data points and smooth the curve, thus effectively reducing the vibration and shock of the robotic arm during movement [12,13,14]. Although high-order polynomial interpolation can provide a smoother curve fit, it may lead to the Runge phenomenon with an excessive number of data points. In addition, they are highly sensitive to minor data point variations, potentially leading to numerical instability. Furthermore, Cong et al. [15] proposed the use of fifth-order Non-uniform B-spline curves in the trajectory planning and crafted a multi-objective trajectory optimization model employing the Non-dominated Sorting Genetic Algorithm-II (NSGA2). Regarding the T-shaped speed curve, Ayazbay et al. [16] applied a mini-mum time-impact planning algorithm to optimize the curve, targeting an enhanced dynamic performance and reduced mechanical wear. However, this curve lacked smoothness at the endpoints, necessitating additional processing to ensure the overall continuity and smoothness of the trajectory. In response to these limitations, researchers have proposed S-shaped speed curves [17,18] as trajectory planning curves, which are distinguished by smooth transitions at the velocity inflection points. This results in less impact force and noise compared to the T-shaped acceleration and deceleration phases. S-shaped speed curves are commonly used in applications requiring precise control and minimization of mechanical vibration.

Heuristic algorithms are increasingly employed in trajectory planning and typically address complex optimization problems when it is difficult to find an exact solution or when the cost of computing an exact solution is too high. Intelligent algorithms, such as genetic algorithms (GA), particle swarm optimization (PSO) algorithms, annealing algorithms (SA), whale optimization algorithms (WOA), and grey wolf optimization (GWO) algorithms provide novel solutions for addressing intricate trajectory planning problems. They have the characteristics of flexibility, robustness, and diversity, which can help users find the global optimal solutions. Among them, the PSO algorithm [19], which simulates the social behavior of bird flocks to achieve optimization, is widely used to optimize the use of robotic arms owing to its simplicity, ease of implementation, and strong ability for multi-modal optimization. However, the PSO algorithm cannot avoid problems of parameter sensitivity and convergence acceleration. Therefore, research teams have improved to varying degrees to overcome these shortcomings [20,21,22,23]. Lu and Wang [24] built an optimal trajectory planning method based on a B-spline and a WOA to improve the work efficiency of a robotic arm. However, they did not fully compare the performance differences between the WOA and other existing algorithms. By combining the back propagation neural network (BPNN) with the GA, the overall performance of the upper limb rehabilitation robotic arm trajectory planning was improved to some extent. However, the parameter adjustment after the combination was very complex and required a large number of experiments and professional knowledge for correction [25]. The SO algorithm excels outstanding in the realm of optimization owing to its exceptional flexibility and robustness. It is not only easy to integrate chaos theory and learning mechanisms to bolster search efficiency but also to maintain efficient exploration ability under poor initial conditions. In addition, it can rapidly adapt to the changes in dynamic environments and optimize search strategies in real time, thereby safeguarding the continuity and effectiveness of the optimization process. Li et al. used a SO algorithm to find optimal paths in dynamic environments. Although its performance has been improved compared with other intelligent algorithms, the improved algorithm still requires careful parameter tuning [26]. To address the shortcomings of the SO algorithm, recent literature [27,28] begun to focus on solving its core mechanism to raise the overall performance of optimization problems. The ISO has also been gradually applied to path planning and to solve general constrained optimization problems.

Therefore, considering the shortcomings of the afore-mentioned studies and the main factors that should be prioritized during the actual operation of the wafer manipulator, this study proposes an improved control strategy for a manipulator that combines the single-segment start–stop S-shaped speed of the circular arc transition, an ISO algorithm, and a time-impact optimization method of trajectory planning. It is worth noting that it has more advantages in terms of the stability of the control effect or the convergence ability of the algorithm compared with other types of methods. And it can adapt to parameter optimization in different scenarios. In summary, the contributions of this study are as follows.

(1)In the area of intelligent optimization methods, the following improvements have been made to the SO algorithm.
The Tent Chaotic Map initializes populations, instead of randomly generating populations with ‘*rand*’ functions.Dynamic learning factors, integrated with random perturbations, replace static learning factors.A cosine annealing learning rate scheme is employed for the adaptive updating of individual positions.The Bayesian method is utilized to optimize thresholds, surpassing the reliance on empirical thresholds.(2)In the area of trajectory planning, the following improvements have been made to the S-shaped speed curve.
The interpolated arc transition supplants the abrupt right-angle transition.The continuous single-segment start–stop mechanism supersedes the segmented start–stop mechanism.

## 2. Materials and Methods

### 2.1. The Structure Parameters of the Wafer-Handling Robotic Arms

Assuming that the task requirements can be realized in a plane, the system model is simplified when the robotic arm transports the wafer, as shown in Figure 1. Before planning the trajectory, it is necessary to establish the link coordinate system correctly. In general, the points along the *Z_i_* axis are aligned with the i-th joint axis. Furthermore, the points along the *X_i_* axis are aligned with a common vertical line. In addition, the *Y_i_* axis is oriented according to the right-hand rule. In this study, the parameters of the linkage rods are expressed using the improved Denavit–Hartenberg (DH) method. The parameters of each joint and link are obtained using the link coordinate system as a reference, as shown in Table 1.

To facilitate the visual analysis of the subsequent trajectory planning and verify the correctness of the simplified model, this study utilizes the MATLAB robot toolbox developed by Peter Corke. It is a powerful robotic simulation and analysis tool that provides a range of capabilities for modeling, simulating, controlling, and visualizing robots. A robotic arm consistent with the simplified model is successfully constructed on the interface by following the building instructions in the MATLAB robot toolbox, as shown in Figure 2.

### 2.2. The Mathematical Model of Multiple Objective Function

In recent years, the optimization of time, energy, impact, and other indicators has been shown to improve the operational efficiency and motion performance of robotic arms in the field of robot trajectory planning [29,30,31]. In the preprocessing stage of trajectory optimization analysis, expected trajectories as input parameters need to be predefined and generated. Therefore, this study will build the expected trajectory through the MATLAB platform, and the specific implementation process will be elaborated in the following sections. Considering the production background and optimization effect, a multi-segment point-to-point trajectory in Cartesian space is adopted to achieve multi-objective optimization of time and impact. In the following section, we will systematically analyze the mathematical model and its optimization process.

Initially, the desired trajectory of the robotic arm is discretized into multiple interpolation points within the Cartesian space. These interpolation points are selected at regular time intervals along the desired trajectory, thereby accurately capturing the positional information of the trajectory at different time instances. Subsequently, the Cartesian coordinates of these interpolation points are stored in the joint controller. By employing the inverse kinematics equations, the Cartesian coordinates of the interpolation points can be further transformed into the corresponding joint angles in the joint space. However, the inverse kinematics equations typically yield two distinct solutions. Therefore, it is necessary to impose geometric constraints within the code to select the unique solution that conforms to the desired trajectory. Once the unique solution is obtained, it is substituted into the robot’s forward kinematics model to derive the Cartesian coordinates of the desired trajectory. Ultimately, by combining the trajectory function with the Cartesian coordinates of the interpolation points, the desired trajectory is precisely reconstructed. It is assumed that the time at which the manipulator reaches each interpolation point is *t_i_*. The time interval between adjacent interpolation points is *h_i_* = *t_i+_*_1_ − *t_i_*, where *i* is the index of the interpolation points (*i* = 0, 1, … *n*). The time interval taken in the code of this paper is 0.002 s, that is, the selection basis of interpolation points. At the same time, through the preliminary analysis, it is clearly observed that there is an inverse correlation between time and impact. When the time of the robotic arm increases, the impact decreases. Therefore, it is impossible to simultaneously obtain the minimum values. Thus, considering the constraints of the geometric joints of the robotic arm, velocity, acceleration, and jerk, the primary goal of this study is to achieve an optimal balance between time and impact. Therefore, a Pareto optimal method is adopted in this study to achieve the tradeoff between optimization objectives. The time function is given by Equation (1). Because the impact quantity can not be directly measured, it is quantified by the acceleration, as shown in Equation (2). The constraints are expressed in Equation (3). The multi-objective optimization function is shown in Equation (4).(1)$$\min F_{1}=\sum_{i=0}^{n}h_{i}$$(2)$$\min F_{2}=\sum_{i=0}^{n}\sqrt{\frac{\int_{0}^{t_{i}}j_{i}^{2}dt}{t_{i}}}$$(3)$$\left\{\begin{array}{l} \left|\theta_{i}\right|\leq \theta_{\max}\\ \left|v_{i}\right|\leq v_{\max}\\ \left|a_{i}\right|\leq a_{\max}\\ \left|j_{i}\right|\leq j_{\max} \end{array}\right.$$(4)$$\min F=w_{1}\cdot \left|F_{1}-F_{1\min}\right|+w_{2}\cdot \left|F_{2}-F_{2\min}\right|$$
where *F*_1_, *F*_1*min*_, and *w*_1_ are the objective function, minimum value, and weight value of time, respectively. *F*_2_, *F*_2*min*_, and *w*_2_ are the objective function, minimum value, and weight of impact, respectively. *h_i_* is the time interval between the *i*-th and *i* + 1-th interpolation point. *t_i_* and *j_i_* denote the time node and jerk value of the *i*-th interpolation point. *θ_max_*, *v_max_*, *a_max_*, and *j_max_* are the maximum rotational angle, speed, acceleration, and jerk of the joint of the manipulator arm, respectively.

### 2.3. The Construction of Trajectory Planning

#### 2.3.1. The Explanation of S-Shaped Speed Curve

The T-shaped speed curve is a widely used method for robotic arm control. Because of its simplicity and ease of control, it is extremely common in industrial applications. As shown in Figure 3, this curve may produce large impacts and vibrations at points A, B, C, and D, potentially causing the wafer to break. By contrast, the S-shaped speed curve uses nonlinear adjustments to increase and decrease the velocity, making the motion process smoother and effectively solving the discontinuity problem of the acceleration of the T-shaped speed curve. Therefore, the S-shaped speed curve is more suitable in scenarios requiring higher operational stability.

The S-shaped speed curve partitions the entire motion process into seven distinct phases. Equation (5) describes the motion characteristics at different stages.(5)$$v(t)=\left\{\begin{array}{ll} v_{s}+\frac{1}{2}j_{\max}\tau_{1}^{2} & 0\leq t\leq t_{1}\\ v_{1}+a_{\max}\tau_{2} & t_{1}\leq t\leq t_{2}\\ v_{2}+a_{\max}\tau_{3}-\frac{1}{2}j_{\max}\tau_{3}^{2} & t_{2}\leq t\leq t_{3}\;\\ v_{3} & t_{3}\leq t\leq t_{4}\\ v_{4}-\frac{1}{2}j_{\max}\tau_{5}^{2} & t_{4}\leq t\leq t_{5}\\ v_{5}-a_{\max}\tau_{6} & t_{5}\leq t\leq t_{6}\\ v_{6}-a_{\max}\tau_{7}+\frac{1}{2}j_{\max}\tau_{7}^{2} & t_{6}\leq t\leq t_{7} \end{array}\right.$$
where *t_k_* (*k* = 0, 1, …7) represents the transition point time of part *k*, which indicates the specific point in time at which each stage begins or ends. *τ_k_* (*k* = 0, 1, …7) represents the local time period of part *k*, which refers to the difference from the starting point of the current part, called time zero, to the time point of the current part. The equation used *τ_k_* = *t_k_ − t_k−_*_1_ (*k* = 1, 2, …7). *v_k_* (*k* = 1, 2, …6) represents the final speed of part *k* and vs. represents the initial speed. *a_max_* is the maximum acceleration. *j_max_* is the maximum acceleration.

#### 2.3.2. The Improvement of the S-Shaped Speed Curve

Trajectory planning is a crucial step in the robotic motion control. When the wafer-handling robotic arm is in operation, it frequently encounters turning points where the direction of movement requires alteration. These turning points often serve as a source of abrupt jerk changes. To alleviate this problem, an S-shaped speed curve adopts a circular arc transition strategy to guarantee a smooth trajectory transition. This strategy not only mitigates the impact and vibration induced by the robotic arm but also contributes to shortening the execution time of the motion trajectory. Prior to constructing the S-shaped speed curve, it is imperative to determine the key interpolation points of the trajectory in Cartesian space, as indicated in Table 2. Simultaneously, the principal motion parameters must be set as presented in Table 3. *v_ p*_0_, *v_ p*_1_, and *v_ p*_2_ are the velocities of the initial point, the circular arc transition section, and the terminal point, respectively. *v_max_*, *a_max_*, and *j_max_* denote the maximum velocity, the maximum acceleration, and the maximum rate of change in the acceleration of an end effector in Cartesian space, respectively. *r* is the radius of the circular arc transition at the inflection point.

The trajectory curve shown in Figure 4. can be generated by applying the parameters listed in Table 3. to the S-shaped speed curve model. In this figure, the black lines represent the direct path without a circular transition, whereas the blue lines show a smooth path with a circular transition. It is noteworthy that the radius of the circular transition can be adjusted according to specific work requirements. Overall, the circular transition not only improves the smoothness and efficiency of the robotic arm’s motion but also helps to protect the robotic arm and its payload, while enhancing the overall operational performance. This transition method allows the robotic arm to perform complex tasks more accurately and reliably.

By analyzing the data in Table 2, it is clear that this trajectory planning task involves three key nodes that the robotic arm must reach in sequence. S-shaped speed curve planning is a standard point-to-point start–stop motion scheme. Specifically, this task is represented by two movements from the starting point to the middle point and then from the middle point to the end point, with zero starting and ending speeds for each movement. Based on the data in Table 3, the velocity image of the scheme is shown in Figure 5. However, this scheme causes the manipulator to frequently start and stop, which not only increases the vibration during the motion process but also leads to time wastage in the acceleration and deceleration stages. To address this issue, an improved scheme is proposed in this study for the single-segment start–stop mechanism, optimizing the starting and stopping parts of the intermediate process into a smooth motion mode with constant speeds at both ends, as shown in Figure 6. When the velocity of the first segment of motion attains the maximum value of the set parameters, instead of reducing the velocity back to zero as in the standard scheme, the second segment of motion proceeds in a uniform transition manner. The specific parameters still refer to Table 3. This improved scheme effectively mitigates the limitations inherent in the conventional approach. Simultaneously, the movement efficiency can be improved and the vibration can be reduced.

Comparing Figure 5 and Figure 6, it can be seen that the improved motion scheme results in a smoother speed curve. Concurrently, it effectively reduces the vibration generated by the motor during starting and stopping. With the basic parameters unchanged, the unoptimized time schedule which does not include arc adjustment and smooth transition took 7.26 s, whereas the optimized time schedule, which includes arc adjustment and smooth transition, is shortened to 5.51 s. After the two parts are optimized, the total time is saved by 1.75 s, a 24.1% reduction compared to the original scheme.

### 2.4. Improved Snake Optimization Algorithm

#### 2.4.1. Initialization Population of a Tent Chaotic Map

The snake optimization (SO) algorithm is an intelligent optimization technique that simulates the specific mating behavior of snakes. It achieves an effective balance between global and local searches by using a unique search mechanism. In the initialization phase of the algorithm, individuals in the population are equally divided into male and female individuals. The composition of this initial population can be represented by Equation (6), where the numbers of male and female individuals are represented by Equations (7) and (8), respectively.(6)$$X_{i}=lb+rand\times (ub-lb)$$(7)$$N_{m}=\frac{N}{2}$$(8)$$N_{f}=N-N_{m}$$
where *X_i_* denotes the position of the *i*-th individual. The random function ‘*rand*’ generates a real random number between 0 and 1. The upper and lower bounds of *X_i_* are *ub* and *lb*, respectively. *N* is the total population size. *N_m_* and *N_f_* are the numbers of individual males and females, respectively.

However, the use of the function ‘*rand*’ in Equation (6) of the SO algorithm may result in an uneven distribution of random numbers. Moreover, because this function is based on a deterministic algorithm, the same random sequence is generated when the same seed is given. This situation may lead to the premature convergence of the heuristic algorithm to the local optimal solution, making it challenging to find the global optimal solution. To address this issue in population initialization, this study introduces a Tent Chaotic Map. It has excellent chaotic characteristics, which can effectively improve the uniformity of population distribution. In addition, compared with Sine and Logistic chaotic map [32,33], its implementation is more intuitive and simple, while Sine and Logistic chaotic map’s late depth search ability is weak, which may cause the algorithm to fall into local optimal. At the same time, the distribution of their output sequence may be uneven under certain parameter values. As a consequence, the Tent Chaos map is a good choice in this study. And the Tent Chaos Map can be expressed as Equation (9). And ‘*x_i_*’ replaces ‘*rand*’ as shown in Equation (10).(9)$$x_{i+1}=\left\{\begin{array}{ll} \frac{x_{i}}{a}, & 0\leq x_{i}<a\\ \frac{1-x_{i}}{1-a}, & a\leq x_{i}\leq 1 \end{array}\right.$$(10)$$X_{i}=lb+x_{i}\times (ub-lb)$$
where *x_i_* and *x_i+_*_1_ stand for the *i*-th and *i* + 1-th values of Tent Chaotic Map, respectively. *X_i_* stands for the *i*-th individual position. Moreover, parameter *a* is the variable value, which is 0.5 in this study [34].

#### 2.4.2. Dynamic Learning Factor with Random Noise

In heuristic algorithms, learning factors play a key role in regulating the search behavior, including the balance between exploration and exploitation. Exploration refers to the ability to search for new solutions randomly, whereas development refers to the ability to search locally for the best solution. The learning factors of the SO algorithm include the constant value *C*_1_ of the food calculation equation, constant value *C*_2_ of the food exploration stage, and constant value *C*_3_ of the food development stage. The initial values of these constants are typically set to *C*_1_ = 0.5, *C*_2_ = 0.05, and *C*_3_ = 2. The equations containing *C*_1_, *C*_2_, and *C*_3_ are Equations (11)–(15). The subscript ‘*m*’ and ‘*f*’ refers to the male and female snakes, respectively. The special mention of female snake expressions will be omitted in the following text.(11)$$Q=C_{1}\times (\frac{t-T}{T})$$(12)$$X_{i,\;m\_1}(t+1)=X_{rand,\;m\_1}(t)\pm C_{2}\times A_{m}\times \left[(ub-lb)\times rand+lb\right]\;$$(13)$$X_{i,\;m\_2}(t+1)=X_{\mathrm{food}}(t)\pm C_{3}\times \mathrm{Temp}\times rand\times \left[X_{\mathrm{food}}(t)-X_{i,\;m}(t)\right]$$(14)$$X_{i,\;m\_3}(t+1)=X_{i,\;m\_3}(t)\pm C_{3}\times F_{m}\times rand\times \left[Q\times X_{best,\;f}-X_{i,\;m}(t)\right]$$(15)$$X_{i,\;m\_4}(t+1)=X_{i,\;m\_4}(t)\pm C_{3}\times M_{m}\times rand\times \left[Q\times X_{i,\;f}(t)-X_{i,\;m}(t)\right]$$
where *Q* is the amount of food consumed. *t* denotes the current number of iterations. *T* denotes the maximum number of iterations. *Temp* is the iteration temperature. *X_i_*_, *m*_1_, *X_i_*_,_
*_m__*_2_, *X_i_*_, *m*_3_, and *X_i_*_, *m*_4_ are the locations of food in the exploration phase of male snakes, food in the development phase of male snakes, combat mode in the development phase of male snakes, and mating mode in the development phase of male snakes, respectively. *X_rand_*_, *m*_1_ is the randomly selected male population. *X_best_f_* is the best position for the females in this population. *A_m_*, *F_m_*, and *M_m_* are measured for the male’s ability to find food, the male’s ability to fight, and the male’s ability to mate.

Because the learning factors *C*_1_, *C*_2_, and *C*_3_ remain unchanged during the iteration, the algorithm may not adapt to the dynamic changes in the solution space. Hence, the delta value is introduced for the normal distribution of random tubs. This perturbation helps the SO algorithm eliminate the trap of local optimal solutions, thus enhancing its global search abilities. In addition, the literature [35] introduces a learning strategy employed within the ISO framework, which centers on acquiring knowledge from the experiences of other group members. This approach aims to mitigate the likelihood of converging on local optima. Nonetheless, the implementation of such specialized learning mechanisms not only entails a notable increase in algorithmic complexity but also exhibits diminished efficacy when adapting to the diverse phases of the optimization procedure. However, the SO algorithm can achieve an effective dynamic balance between global and local searches by dynamically adjusting these fixed learning factors *C*_1_, *C*_2_, and *C*_3_, thereby improving the performance and efficiency of the algorithm. The concrete expression of the random disturbance is shown in Equations (16) and (17). In Equation (19), *alpha* is used to store the parameter values for learning factors *C*_1_, *C*_2_, and *C*_3_.(16)delta=sigma×randn(17)alpha(t+1)=alpha(t)+delta
where *sigma* represents the amplitude of the perturbation. It is 0.1 in this paper [36]. *alpha* is limited to a range between 0.01 and 1. Random function ‘*randn*’ represents a random value with a standard normal distribution.

#### 2.4.3. Cosine Annealing Learning Rate

In Equations (12)–(15), the strategy for updating the individual positions is relatively simple. Without considering adjusting the strategy as an iteration progress, the SO algorithm mainly relies on the current individual’s position and velocity vector to determine the next individual’s position. This method is more likely to cause the algorithm to become stuck in a locally optimal solution. To improve the update strategy of the original SO algorithm, this study introduced a cosine learning annealing rate. This strategy simulates the dynamic adjustment of the learning rate based on the cosine function, which can dynamically adjust the speed of the position updating based on the number of iterations. In the early stages of the iteration, the learning rate can be reduced rapidly to achieve fast convergence. In the later stages of the iteration, fine adjustments are made to optimize the precision of the solution by reducing the learning rate, as shown in Figure 7. The specific expression of the cosine learning annealing rate is shown in Equation (18).(18)$$\eta_{t}=\eta_{\min}+\frac{1}{2}(\eta_{\max}-\eta_{\min})\times (1+\cos(\frac{T_{cur}}{T_{\max}}\pi ))$$
where *η**_t_* is the current learning rate. *η_min_* and *η_max_* are the minimum and maximum values of the learning rate, which are 0.001 and 0.01 in this paper. *T_cur_* and *T_max_* are the current and total iterations, respectively. *T_max_* is usually set to half the number of iterations of the main algorithm, which is 20 in this article [37].

By comparison with the cosine annealing learning rate, Gaussian adaptive parameters [38] are sensitive to initial conditions, prone to premature convergence, and may introduce additional computational complexity. However, the cosine annealing learning rate exhibits periodic change, which enables the SO algorithm to adopt diverse search strategies in different stages, thus enhancing the dynamic adjustment ability of the algorithm. This flexibility helps the algorithm to avoid premature convergence during the search process, thereby increasing the possibility of finding a global optimal solution. Therefore, by introducing the cosine annealing learning rate based on Equations (12)–(15), the improved Equations (19)–(22) are obtained.(19)$$X_{i,\;m\_1}(t+1)=\eta_{t}\times X_{rand,\;m\_1}(t)\pm \eta_{t}\times C_{2}\times A_{m}\times \left[(ub-lb)\times rand+lb\right]\;$$(20)$$X_{i,\;m\_2}(t+1)=\eta_{t}\times X_{\mathrm{food}}(t)\pm \eta_{t}\times C_{3}\times \mathrm{Temp}\times rand\times \left[X_{\mathrm{food}}(t)-X_{i,\;m}(t)\right]$$(21)$$X_{i,\;m\_3}(t+1)=\eta_{t}\times X_{i,\;m\_3}(t)\pm \eta_{t}\times C_{3}\times F_{m}\times rand\times \left[Q\times X_{best,\;f}-X_{i,\;m}(t)\right]$$(22)$$X_{i,\;m\_3}(t+1)=\eta_{t}\times X_{i,\;m\_3}(t)\pm \eta_{t}\times C_{3}\times F_{m}\times rand\times \left[Q\times X_{best,\;f}-X_{i,\;m}(t)\right]$$

#### 2.4.4. The Bayesian Method to Find the Optimization Threshold

Bayesian optimization can be applied not only to neural network architectures based on snake algorithms to improve system performance [39], but also to optimize hyperparameter combinations in machine learning. In the process of determining threshold values of the SO algorithm, parameter setting often depends on the experience and intuition of algorithm developers or needs to be determined by practical users through practice. This approach lacks universality and requires considerable time for experimentation and adjustments. If the parameters are not selected properly, the adaptability of the algorithm may be affected. To solve the problem in which the SO algorithm has difficulty adapting parameters in the iterative process, this study introduces a hyperparameter-tuning method commonly used in machine learning, which is known as Bayesian optimization. The Gaussian process mainly includes Gaussian process regression and an acquisition function.

First, it establishes prior knowledge of the global behavior of the objective function through a Gaussian process. The mean vector and covariance matrix involved in the prior distribution are shown in Equations (23) and (24). Subsequently, using the observed objective function to obtain values at different input points, this prior knowledge is updated to form a posterior distribution. The posterior mean and variance are given by Equations (25) and (26). Then, the improved collection function is expected to consider the current best observations and the posterior distribution of the Gaussian process to determine the next evaluation point. The collection function used in this study is expected to improve equation shown in (27). In short, this method of guiding the search based on existing knowledge can avoid many useless attempts.(23)$$\mu_{0}={\left[\mu_{0}(x_{1}),\ldots ,\mu_{0}(x_{k})\right]}^{T}$$(24)$$\sum_{0}=\left[\begin{array}{ccc} \sum_{0}(x_{1},x_{1}) & \cdots & \sum_{0}(x_{1},x_{k})\\ \vdots & \ddots & \vdots \\ \sum_{0}(x_{k},x_{1}) & \cdots & \sum_{0}(x_{k},x_{k}) \end{array}\right]$$

We assume that the above expression follows a multi-variate normal distribution, where *μ*_0_(*x*) represents the mean function. ∑_0_(*x*,*x*’) represents the kernel function, also known as the covariance function, which describes the similarity between points *x* and *x*’ in the input space.(25)$$\mu_{n}(x)=\mu_{0}(x)+\sum_{0}\left(x,x_{1}:n\right)\times {\left(K_{x_{1}:n,x_{1}:n}+\sigma^{2}I\right)}^{-1}\times (f(x_{1}:n)-\mu_{0}(x_{1}:n))$$(26)$$\sigma_{n}^{2}(x)=\sum_{0}(x,x)-\sum_{0}\left(x,x_{1}:n\right)\times {\left(K_{x_{1}:n,x_{1}:n}+\sigma^{2}I\right)}^{-1}\times \sum_{0}\left(x_{1},n:x\right)$$(27)$$EI(x)=E\left[\max\left(f(x)-f^{*},0\right)\right]$$
where *K_x_*_1:*n*,*x*1:*n*_ is the kernel matrix. *σ*^2^ is the noise variance. *f*(*x*_1_:*n*) is the observed vector. *f** = *max_i_* = 1, … *n* is the currently observed best target value. *f*(*x_i_*) is the function value.

This study uses the ‘*bayesopt*’ function in MATLAB 2022B, which is an easy to use and flexible Bayesian optimization tool. With the help of this tool, three key threshold parameters (food threshold, temperature threshold and mating threshold) that affect the balance between exploration and development in SO algorithm are mainly optimized. The original parameter settings are 0.25, 0.6, and 0.6, respectively. Under an integrated Bayesian optimization framework, they are redefined as three adjustable variables: threshold_1_, threshold_2_, and threshold_3_. The construction of the parameter space is based on the original thresholds, extending ±0.2 on both sides, ensuring the validity of the parameters while avoiding the computational burden caused by an overly large search space. Considering the characteristics of the low-dimensional parameter space, a Bayesian optimization configuration with a maximum of 10 iterations is adopted in this study, which not only guarantees optimization efficiency but also effectively saves computational resources.

The method establishes a cooperative mechanism between Bayesian optimization framework based on Gaussian process and SO objective function, and realizes intelligent optimization of optimal threshold combinations through parameter search strategy guided by this model. To improve parameter generalization, K-fold cross-validation is employed for robustness assessment. This approach partitions the dataset into K subsets, analyzing performance variations across these subsets to effectively prevent overfitting and guide optimization toward parameter configurations with enhanced generalization capabilities. This safeguard mechanism not only enhances the reliability of parameter optimization but also provides theoretical support for the stable performance of the algorithm in unknown scenarios.

#### 2.4.5. ISO Algorithm Steps

The flow chart of the ISO algorithm, which illustrates the step-by-step process, is shown in Figure 8.

The ISO algorithm is outlined in Algorithm, as shown in Table 4.

## 3. Results

### 3.1. Ability Test and Analysis for ISO

#### 3.1.1. Parameter Settings and Simulation Results

In this study, all algorithm experiments and simulations are performed using MATLAB 2022B software platform. To verify the convergence performance and accuracy of the ISO algorithm, five heuristic algorithms—SO [40], SA [41], PSO [19], GWO [42], and WOA [43]—are added for comparison. In addition, to ensure the rigor and accuracy of the experimental results, the CEC2005 benchmark test set is selected as the objective function, as listed in Table 5. Five unimodal functions (*F*_1_–*F*_5_) and three multi-modal functions (*F*_6_–*F*_8_) are listed in Table 5. Through an in-depth analysis of these benchmark functions, the performance of the ISO algorithm is evaluated to a certain extent.

Before simulating the benchmark test functions for the six heuristic algorithms, a brief introduction to their main parameter settings is necessary to ensure that researchers can obtain similar experimental results when conducting simulations. To ensure the fairness of experiments, the population size (*N*) of all algorithms is set to 50, the maximum iteration number (*T_max_*) is set to 100, the dimension (*Dim*) is set to 30, and each algorithm is run 20 times independently. For the other key parameters within the algorithms, recommended values based on experience are used for setting, and the specific parameter values are listed in Table 6. After successfully configuring these parameters, a MATLAB script is used to start the six algorithms simultaneously. The experimental results are summarized in Table 7, and the iteration curves of the benchmark test functions are displayed in Figure 9.

#### 3.1.2. Simulation Result Analysis

The results of the eight benchmark functions listed in Table 7 are comprehensively analyzed and compared. As shown in Figure 9a–e, it is clearly observed that when the dimension (*Dim*) is 30, the ISO algorithm achieves an optimal fitness value of 0 for the *F*_1_ and *F*_3_ problems in the unimodal test function. Its convergence curve can rapidly jump out and continue to decline after falling into the local optimum for a short time. This indicates that the ISO algorithm can effectively escape from a local optimal solution and successfully obtain a global optimal solution. By contrast, to a certain extent, the other five algorithms fall into the local optimum. For the *F*_2_ and *F*_4_ problems, although the average value of the ISO algorithm is not zero, the variance is zero, indicating that the algorithm can obtain a global optimal solution to some extent. Its average precision is better than that of the second ranked GWO algorithm. In addition, the convergence speed of the ISO algorithm is faster than those of the other algorithms. For the *F*_5_ problem, the performances of the ISO algorithm and the other five algorithms are similar, but its overall optimization precision is slightly higher, and the convergence curve is better, without falling into the local optimum too early. Overall, the optimization effect of the ISO algorithm on unimodal functions is significantly better than that of the other five algorithms. Further analyzing Figure 9f–h, when the dimension (*Dim*) is 30 in the multi-modal test functions, the SO, GWO, WOA, and ISO algorithms can achieve the global optimal value in the *F*_6_ and *F*_7_ problems with good overall performance, but the ISO algorithm has a faster iteration speed. However, the performance of the six algorithms is not outstanding in the *F*_8_ problem, but the average value and variance of the ISO algorithm are the best, with an optimization precision approximately twice that of the other algorithms. In summary, the ISO algorithm demonstrates higher optimization precision in both unimodal and multi-modal functions than the other algorithms. It also improves the avoidance of falling into the local optimum compared to the SO algorithm, showing the expected optimization effect.

#### 3.1.3. The Friedman Test

Further analysis of the results from 20 experimental groups using the Friedman test method reveal significant differences in the overall performance of the six algorithms across eight benchmark functions. It is a flexible and effective tool that does not depend on the specific distribution of data, making it particularly suitable for comparing the algorithm performance. When dimension (*Dim*) is set to 30, the average ranking of the mean and variance of each algorithm is shown in Table 8. A lower average ranking indicates that the algorithm performed better, and the results are more stable. According to the data in Table 8, the ISO algorithm achieved the highest rankings in both mean value and variance. Compared with the GWO algorithm, which ranked second, the average ranking score of the ISO algorithm’s mean and variance are 38.9% higher. Combined with the SO algorithm, the ISO algorithm shows a significant improvement in the average ranking score of the mean and variance, reaching 60.8% and 63.4%, respectively. Considering the overall performance of the algorithm, the ISO algorithm exhibits better performance in terms of stability and convergence accuracy.

### 3.2. Time-Jerk Optimal Trajectory Planning Based on ISO

After elaborating on the ISO algorithm and the modified S-shaped speed curve in detail, a combination of the two is performed. The objective function is obtained in this study, as presented in Equation (4). Firstly, Equation (4) adopts a compromise algorithm by adjusting the values of *w*_1_ and *w*_2_ to determine the equilibrium point in the multi-objective optimization. Because the weights of time and impact are considered consistent in this study, both *w*_1_ and *w*_2_ are set to 0.5. Secondly, the time function *F*_1_ and the impact function *F*_2_ belong to different categories of physical quantities. When the weighted sum of their calculation results is taken directly, it leads to an imbalance between the time and impact. To address this issue, a minimum–maximum normalization method is employed in this study. It is typically used to eliminate the influence of dimensions among different features, making the data comparable on the same scale, as shown in Equations (28) and (29).(28)$$F_{1\_norm}=\frac{F_{1}-\min(F_{1})}{\max(F_{1})-\min(F_{1})}$$(29)$$F_{2\_norm}=\frac{F_{2}-\min(F_{2})}{\max(F_{2})-\min(F_{2})}$$

After the above operations, the domains of the values for *F*_1_ and *F*_2_ functions are restricted within the same range. Based on this, a set of excellent solutions is selected by solving a multi-objective function. Then, the optimal solution is identified using the Pareto optimization method. The Pareto judgment standard is as follows. If a new solution improves at least one objective without causing a performance decline in other objectives, then the new solution is considered to be superior to the original solution. When these Pareto optimal solutions are smoothly connected, a Pareto frontier is formed. Ultimately, this study aims to determine a balance point on this curve to achieve the best overall effect.

The specific parameter configuration of the simulation experiment can be found in Table 9 for the application of the ISO algorithm to the S-shaped speed curve optimization problem, and the kinematic parameters are set according to Industrial Robot ISO 9283:1998. In the simulation experiments, *a_max_* and *j_max_* are set as independent variables and their upper bound (*ub*) and lower bound (*lb*) are between 80 and 120. In the SO algorithm, the empirical values of threshold parameters threshold_1_, threshold_2_, and threshold_3_ are 0.25, 0.6, and 0.6, respectively. To determine more reasonable parameter combinations, the ranges of these thresholds are unified and adjusted. The range of threshold_1_ is limited to 0.05 to 0.45, while the ranges of threshold_2_ and threshold_3_ are adjusted to 0.4 to 0.8. Correspondingly, the search range of the threshold parameter (Bayesian threshold) in the Bayesian optimization process is also set to 0.05 to 0.8 to facilitate determining the optimal parameter configuration. The other parameters have the same meanings as those described earlier. The Bayesian optimization threshold simulation results are shown in Table 10, which includes the iteration, the actual observed objective value of the current iteration, the best observed value among all observations (BestSoFar_observed), the model-predicted best value (BestSoFar_estimated), and the best thresholds combination obtained in this generation. However, because there are not enough data to support the construction of the first-generation model of Bayesian optimization, Objective, BestSoFar_estimated, and BestSoFar_observed are not included in the calculation. In addition, a simulation diagram of the minimum objective value and function calculation times corresponding to Bayesian optimization is shown in Figure 10, where the initial value of the first generation is replaced by number 1. For the dual-objective optimization problem considered in this study, the distribution of the Pareto frontier is generally in the form of a curve, which is intuitively shown in Figure 11.

To verify the performance of Bayesian optimization in optimizing the parameters of the objective function, a 5-fold cross-validation method was adopted in this study. The experimental results are shown in Figure 12. It can be clearly seen that the performance of this model during the 5-fold cross-validation process is relatively stable, with a small fluctuation range of the Single Scores, which fully indicates that the model has a certain fitting ability. Generally speaking, a cross-validation score within 0.2 can be regarded as excellent, and a score between 0.2 and 0.5 indicates average performance. The average score of this experiment is approximately 0.4, which indicates that the model has a certain predictive ability on this dataset as a whole, but there is still room for improvement.

According to the data provided in Table 10, Bayesian optimization is used to select the second-generation simulation results with the minimum objective function value. This selection is based on minimization of the target value during the iteration. Figure 10. clearly shows the dynamic change in the target value with the iteration, making the change process clear. Further analysis of Figure 11, the minimum values of *F*_1_ and *F*_2_, respectively, reach 5.6294 s and 115.937 mm/s. And the two minimum values are, respectively, located at the two ends of the Pareto frontier composed of the red dots. To optimize both the time and impact, an optimal advantage must be found on the Pareto frontier. A specific point is identified in Figure 11, which achieves an optimal balance between the time and impact. When *a_max_* and *j_max_* are 78.1476 mm/s^2^ and 119.8756 mm/s^3^, the time and impact of the Pareto optimal value are 5.8028 s and 167.5427 mm/s, respectively.

Using the computed *a_max_* and *j_max_*, these data are reintegrated into the trajectory planning of the S-shaped speed curve. Through this process, it is possible to accurately determine the partial displacement of the trajectory planning in the directions of the *X*-axis and *Y*-axis, as well as the associated combined velocities, combined accelerations, and combined accelerations, as shown in Figure 13. These figures clearly show that the planned S-shaped speed curve exhibited smooth characteristics during movement, effectively avoiding sudden changes and shocks. This is extremely important for precision operations such as wafer handling, as it helps reduce the damage that can be caused by impact.

In addition, this study meticulously records the key parameters, particularly the joint angles, during the S-shaped speed curve movement. Two sets of joint angles consisting of several interpolation points are substituted into the forward kinematics formula, as shown in Equation (30). *X* and *Y* represent the coordinate values in the Cartesian coordinate system, respectively.(30)$$\left\{\begin{array}{l} X=link_{1}\times \cos\theta_{1}+link_{2}\times \left(\cos\theta_{1}+\cos\theta_{2}\right)\\ Y=link_{1}\times \sin\theta_{1}+link_{2}\times \left(\sin\theta_{1}+\sin\theta_{2}\right) \end{array}\right.$$

The actual motion trajectory depicted in the planar space is shown in Figure 14. By comparing Figure 4 and Figure 14, the high consistency of the two trajectories can be clearly verified. This comparison clearly validates the accuracy and reliability of the ISO. This planning method not only elevates the accuracy of trajectory planning, but also ensures the safety and reliability of the motion.

## 4. Conclusions

This study proposes an improved snake optimization (ISO) algorithm. Meanwhile, a single-segment start–stop S-shaped speed curve with arc transition is constructed based on ISO, which is a method specifically designed for the rapid and stable operation requirements of the wafer-handling robotic arm. This method extensively investigates and optimizes on the key issues of time and impact within the context of robotic arm trajectory planning. Through a comparative analysis with the original S-shaped speed curve scheme under the same parameter settings, the new method improves the time efficiency by 24.1%, thereby significantly reducing the time required for the robotic arm to complete the task. Furthermore, the performance of the ISO is compared with that of SO, SA, PSO, GWO, and WOA on a series of benchmark test functions. The results indicate that, in terms of the two key performance indicators, average value and variance, the ISO algorithm outperforms the second ranked GWO algorithm by a margin of 38.9%. These improved simulations fully demonstrate the effectiveness and superiority of the proposed method in addressing the trajectory planning problem for wafer-handling robotic arms.

In this study, the shortcomings of the SO algorithm have been extensively studied and improved. While the proposed enhancements improve global search capabilities, the risk of local optima convergence persists. Additionally, although Bayesian optimization improves the search efficiency, it consumes more computer resources. Therefore, future research will prioritize refining the search mechanism to address these limitations of the SO algorithm. Simultaneously, we plan to explore the combination of Bayesian optimization and bandit-based methods, aiming to reduce computational costs while improving the optimization efficiency.

## Figures and Tables

**Figure 1 sensors-25-01872-f001:**
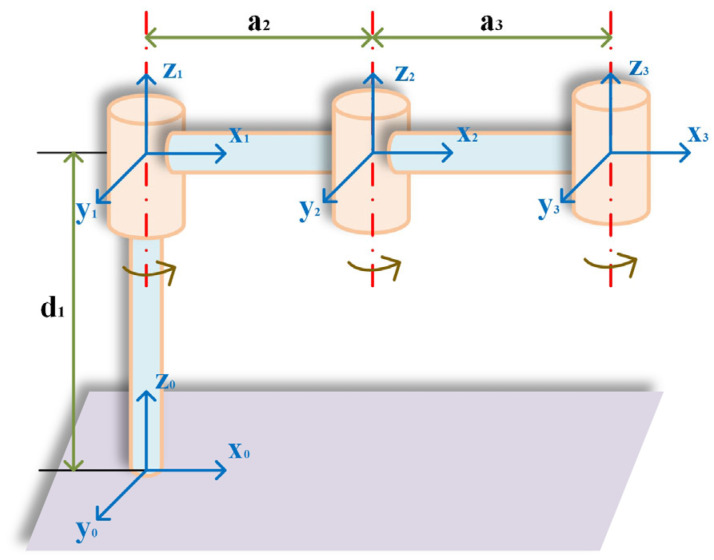
Simplified model of the wafer-handling robotic arm.

**Figure 2 sensors-25-01872-f002:**
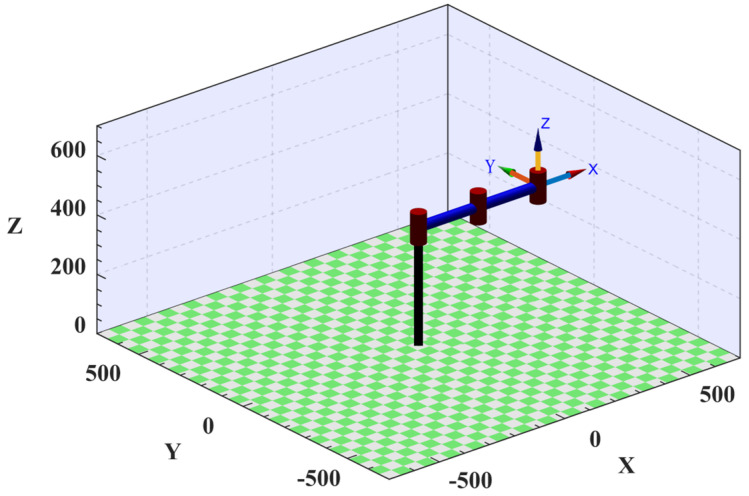
Wafer−handling robotic arm in MATLAB.

**Figure 3 sensors-25-01872-f003:**
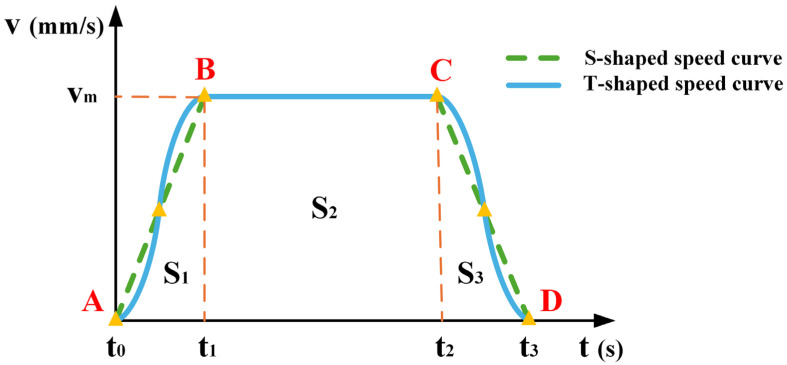
Correlation chart between the T-shaped and S-shaped speed curves.

**Figure 4 sensors-25-01872-f004:**
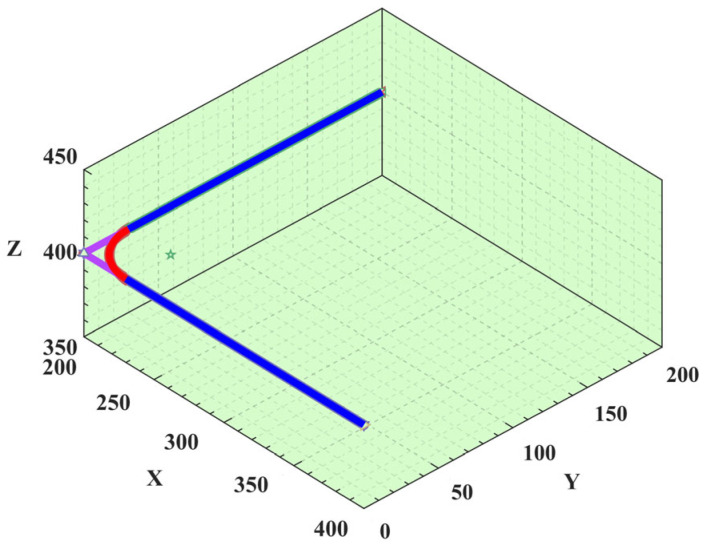
Contrast diagram of arc transition and right-angle path.

**Figure 5 sensors-25-01872-f005:**
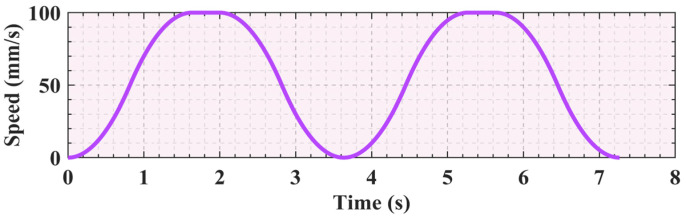
The continuous start–stop S-shaped speed curve.

**Figure 6 sensors-25-01872-f006:**
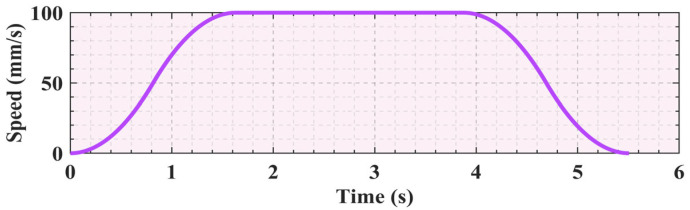
The improved start–stop S-shaped speed curve.

**Figure 7 sensors-25-01872-f007:**
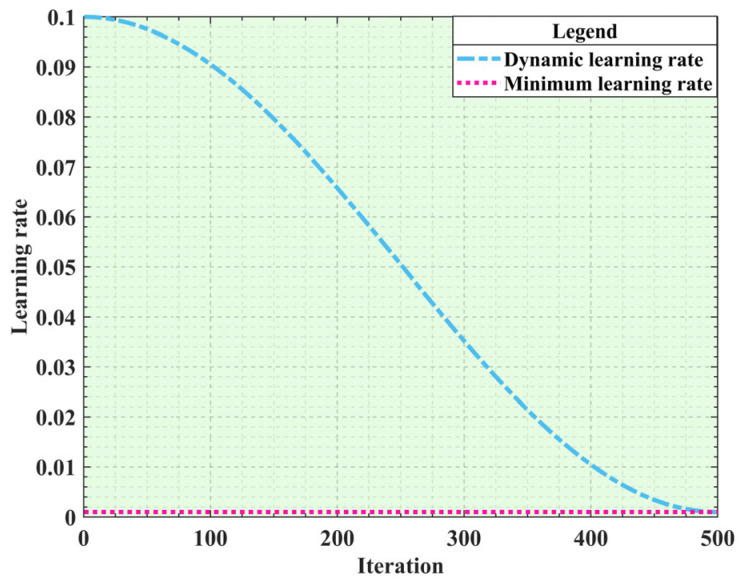
Cosine annealing learning rate iteration graph.

**Figure 8 sensors-25-01872-f008:**
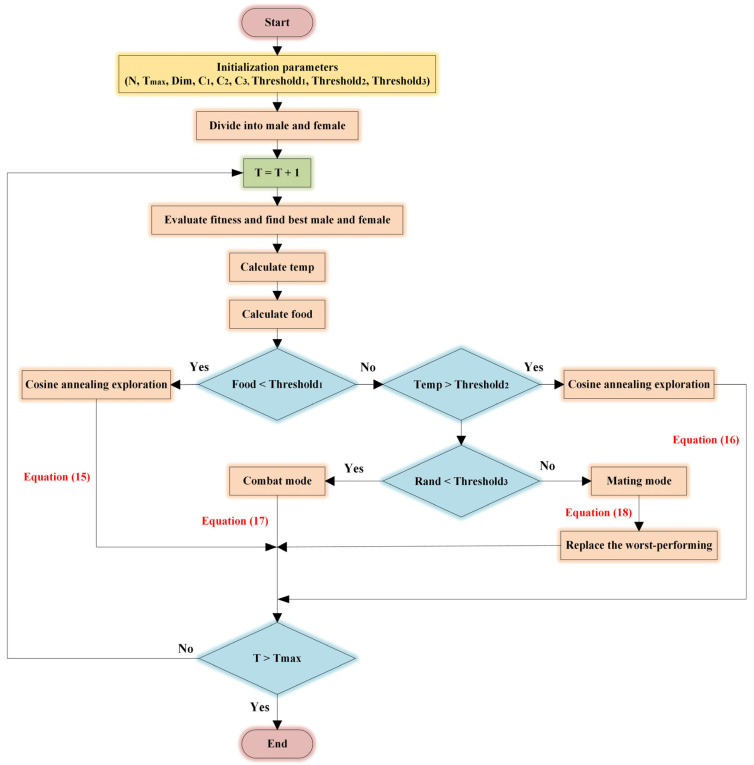
The flow chart of the improved snake optimization algorithm.

**Figure 9 sensors-25-01872-f009:**
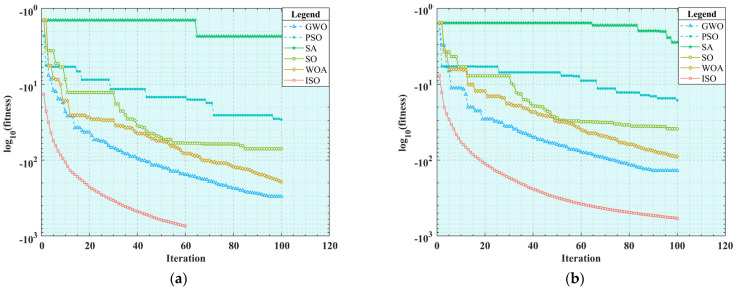
Convergence curve of the reference function: (**a**) *F*_1_; (**b**) *F*_2_; (**c**) *F*_3_; (**d**) *F*_4_; (**e**) *F*_5_; (**f**) *F*_6_; (**g**) *F*_7_; (**h**) *F*_8_.

**Figure 10 sensors-25-01872-f010:**
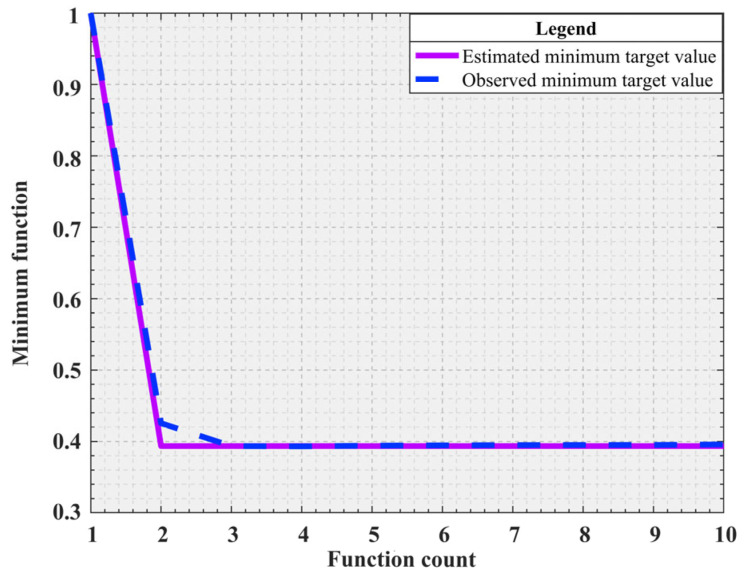
Function count.

**Figure 11 sensors-25-01872-f011:**
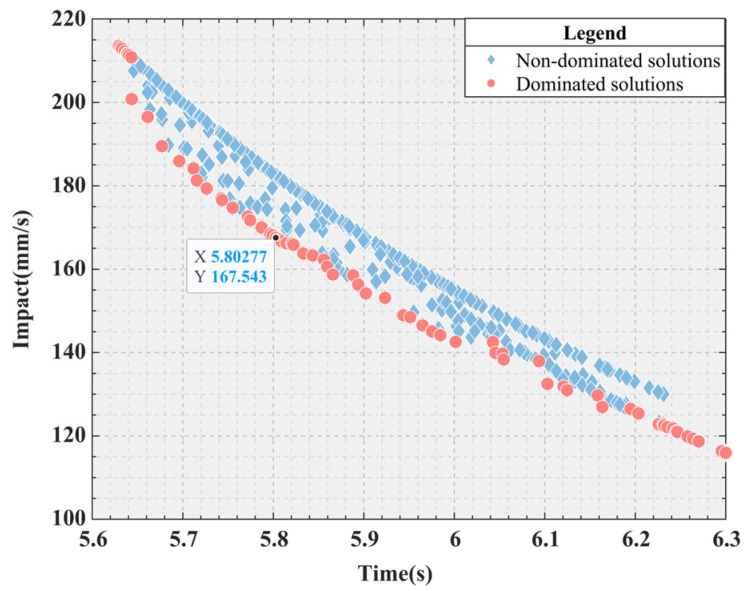
Pareto frontier.

**Figure 12 sensors-25-01872-f012:**
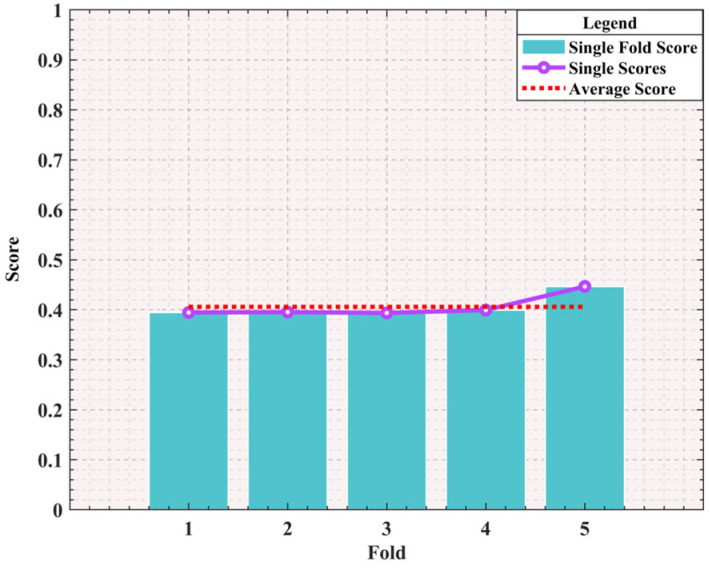
K-Fold Cross Validation Performance (K = 5).

**Figure 13 sensors-25-01872-f013:**
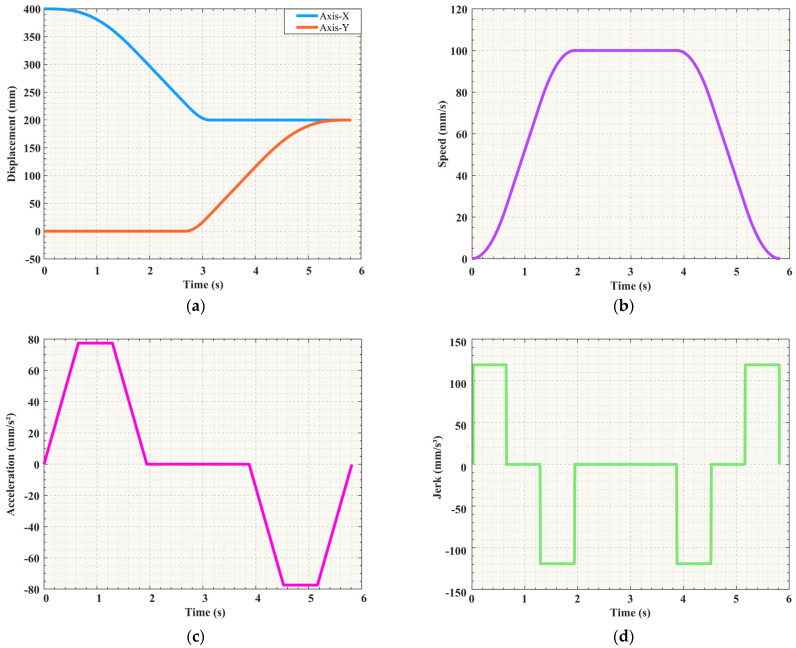
Kinematic image: (**a**) Displacement curve in Cartesian space; (**b**) speed curve in Cartesian space; (**c**) acceleration curve in Cartesian space; (**d**) jerk curve in Cartesian space.

**Figure 14 sensors-25-01872-f014:**
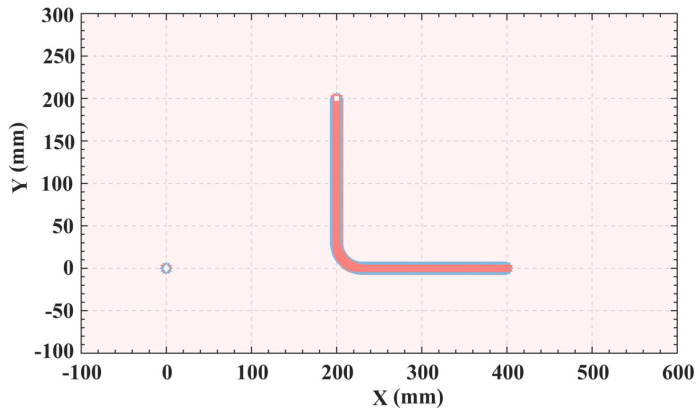
Motion trajectories in a two−dimensional plane.

**Table 1 sensors-25-01872-t001:** Linkage parameters of the 3R robotic arm.

Link*_i_*	*a_i_*/mm	*α_i_*/(°)	*d_i_*/mm	*θ_i_* (/°)	Range of *θ_i_*/(°)
1	0	0	400	0	−120–120
2	0	238.4	0	0	−120–120
3	0	238.4	0	0	−120–120

**Table 2 sensors-25-01872-t002:** Segmentation points in Cartesian space.

Starting Point	Intermediate Point	Ending Point
(400, 0, 400)	(200, 0, 400)	(200, 200, 400)

**Table 3 sensors-25-01872-t003:** The main motion parameters of the S-shaped speed curve.

*v_ p*_0_** (mm/s)	*v_ p*_1_** (mm/s)	*v_ p*_2_** (mm/s)	*v_max_* (mm/s)	*a_max_* (mm/s^2^)	*j_max_* (mm/s^3^)	*r* (mm)
0	100	0	100	100	100	30

**Table 4 sensors-25-01872-t004:** ISO pseudo-code.

1: Initialize parameters 2: Initialize the population using the Tent Chaotic Map as Equation (13) 3: Divide population into *N_m_* and *N_f_* using Equations (10) and (11) 4: while (*T* < *T_max_*) do 5: Evaluate the fitness and identify the best female and male 6: Calculate the iteration temperature and food quantity Equation (14) 7: if (Food < threshold_1_) then 8: Cosine annealing exploration using Equation (15) 9: else if (*Temp* > threshold_2_) then
10: Cosine annealing exploration using Equation (16) 11: else 12: if (*rand* < threshold_3_) then 13: Enter Fight Mode using Equation (17) 14: else 15: Enter Mating Mode using Equation (18) 16: Replace the worst-performing female and male 17: end if 18: end if 19: end while 20: Return the optimal solution

**Table 5 sensors-25-01872-t005:** Test functions.

Function	Range	Min
$F_{1}(x)=\sum_{i=1}^{n}x_{i}^{2}$	[−100,100]	0
$F_{2}(x)=\sum_{i=1}^{n}\left|x_{i}\right|+\prod_{i=1}^{n}\left|x_{i}\right|$	[−10,10]	0
$F_{3}(x)={\sum_{i=1}^{n}\left(\sum_{j=1}^{i}x_{j}\right)}^{2}$	[−100,100]	0
$F_{4}(x)=\max_{i}\left\{\left|x_{i}\right|,1\leq i\leq n\right\}$	[−100,100]	0
$F_{5}(x)=\sum_{i=1}^{n}ix_{i}^{4}+randm[0,1)$	[−128,128]	0
$F_{6}(x)=\sum_{i=1}^{n}\left(x_{i}^{2}-10\cos(2\pi x_{i})+10\right)$	[−5.12,5.12]	0
$F_{7}(x)=\frac{1}{4000}\sum_{i=1}^{d}x_{i}^{2}-\prod_{i=1}^{d}\cos(\frac{x_{i}}{\sqrt{i}})+1$	[−600,600]	0
$\begin{array}{ll} F_{8}(x)= & 0.1\left\{\sin^{2}(3\pi x_{1})+\sum_{i=1}^{n}{\left(x_{i}-1\right)}^{2}\left[1+\sin^{2}(3\pi x_{i}+1)\right]+{\left(x_{n}-1\right)}^{2}\left[1+\sin^{2}(2\pi x_{n})\right]\right\}\\ & +\sum_{i=1}^{n}u(x_{i},5,100,4),\\ u(x_{i},a,k,m)=\left\{\begin{array}{l} k{\left(x_{i}-a\right)}^{m},x_{i}>a\;\\ 0,\;a>x_{i}>-a\\ k{\left(-x_{i}-a\right)}^{m},x_{i}<-a \end{array}\right. \end{array}$	[−50,50]	0

**Table 6 sensors-25-01872-t006:** Parameter settings for the algorithms.

**Algorithm**	**Parameter Settings**
SO	Theshold_1_ = 0.25, Theshold_2_ = 0.6, Theshold_3_ = 0.6, *C*_1_ = 0.5, *C*_2_ = 0.05, *C*_3_ = 2
SA	*Temp* = 100, *Temp_min_* = 0.01, *alpha* = 0.95
PSO	*w* = 0.9, *c*_1_ = 2, *c*_2_ = 2, *V_max_* = 0.3 ∗ *ub*, *V_min_* = 0.3 ∗ *lb*
GWO	*r*_1_ = *r*_2_ = *rand*
WOA	*r*_1_ = *r*_2_ = *rand*, *p* = *rand*, *b* = 1
ISO	*sigma* = 0.1, *alpha_min_* = 0.01, *alpha_max_* = 1.0, *eta_max_* = 0.01, *eta_min_* = 0.001

**Table 7 sensors-25-01872-t007:** Results obtained by different algorithms (*D* = 30).

Function		SO	SA	PSO	GWO	WOA	ISO
*F* _1_	MEAN	8.27 × 10^−40^	7.30 × 10^−2^	3.04 × 10^−12^	6.24 × 10^−105^	8.40 × 10^−79^	0
	STD	2.51 × 10^−39^	2.27 × 10^−1^	6.51 × 10^−12^	2.79 × 10^−104^	2.10 × 10^−78^	0
*F* _2_	MEAN	3.95 × 10^−21^	4.26 × 10^−2^	1.38 × 10^−7^	8.55 × 10^−57^	2.95 × 10^−40^	9.14 × 10^−214^
	STD	5.26 × 10^−21^	2.87 × 10^−2^	2.22 × 10^−7^	3.13 × 10^−56^	7.95 × 10^−40^	0
*F* _3_	MEAN	2.16 × 10^−39^	3.85 × 10^−2^	6.47 × 10^−12^	6.66 × 10^−107^	2.18 × 10^−79^	0
	STD	7.04 × 10^−39^	8.37 × 10^−2^	2.16 × 10^−11^	2.98 × 10^−106^	6.74 × 10^−79^	0
*F* _4_	MEAN	2.01 × 10^−20^	1.34 × 10^−1^	1.38 × 10^−6^	1.27 × 10^−53^	8.56 × 10^−40^	2.27 × 10^−213^
	STD	3.54 × 10^−20^	9.71 × 10^−2^	2.36 × 10^−6^	3.93 × 10^−53^	2.64 × 10^−39^	0
*F* _5_	MEAN	7.01 × 10^−4^	6.64 × 10^−1^	1.77 × 10^−3^	5.21 × 10^−4^	7.33 × 10^−4^	1.75 × 10^−4^
	STD	7.23 × 10^−4^	8.91 × 10^−1^	1.26 × 10^−3^	5.08 × 10^−4^	7.05 × 10^−4^	2.72 × 10^−4^
*F* _6_	MEAN	0	1.79 × 10^−2^	2.39 × 10^−12^	0	0	0
	STD	0	3.42 × 10^−2^	3.35 × 10^−12^	0	0	0
*F* _7_	MEAN	0	1.67 × 10^−2^	6.15 × 10^−12^	0	0	0
	STD	0	3.42 × 10^−2^	3.35 × 10^−12^	0	0	0
*F* _8_	MEAN	1.74 × 10^1^	1.10 × 10^9^	4.41 × 10^1^	1.24 × 10^0^	8.13 × 10^−1^	2.06 × 10^−1^
	STD	3.32 × 10^1^	1.35 × 10^8^	2.59 × 10^1^	4.26 × 10^−1^	3.08 × 10^−1^	2.03 × 10^−1^

**Table 8 sensors-25-01872-t008:** The Friedman test table.

Algorithm	SO	SA	PSO	GWO	WOA	ISO
Mean rank	3.5	6	5	2.25	2.875	1.375
STD rank	3.75	6	4.875	2.25	2.75	1.375

**Table 9 sensors-25-01872-t009:** The main parameter setting.

*v_ p*_0_** (mm/s)	*v_ p*_1_** (mm/s)	*v_ p*_2_** (mm/s)	*v_max_* (mm/s)	*r*_1_ (mm)	Bayesian Evaluations	Bayesian Threshold	*T_max_*	*Dim*	*N*	*lb*	*ub*
0	100	0	100	30	10	[0.05,0.8]	100	2	20	[80,120]	[80,120]

**Table 10 sensors-25-01872-t010:** The iterative graph of Bayesian thresholds.

Iteration	Objective	BestSoFar (Observed)	BestSoFar (Estimated)	Threshold_1_	Threshold_2_	Threshold_3_
1	—	—	—	0.4227	0.7616	0.5225
2	0.3936	0.3936	0.4255	0.0989	0.7764	0.5705
3	0.3952	0.3936	0.3936	0.1721	0.7210	0.7933
4	0.3969	0.3936	0.3932	0.0752	0.4968	0.5437
5	0.3953	0.3936	0.3942	0.1403	0.6057	0.7633
6	0.3991	0.3936	0.3946	0.0500	0.5764	0.5441
7	0.3957	0.3936	0.3949	0.1485	0.5278	0.4003
8	0.3969	0.3936	0.3951	0.1113	0.7425	0.4637
9	0.3955	0.3936	0.3951	0.1592	0.7106	0.7313
10	0.3979	0.3936	0.3958	0.1587	0.6833	0.6341

## Data Availability

Data sharing not applicable. No new data were created or analyzed in this study. Data sharing is not applicable to this article.

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
