# Peer review of "Time-Impact Optimal Trajectory Planning for Wafer-Handling Robotic Arms Based on the Improved Snake Optimization Algorithm"

_sensors, 2025, doi:10.3390/s25061872_

Round 1
Reviewer 1 Report
Comments and Suggestions for Authors
The outstanding feature of the manuscript is the improved snake optimization (ISO) approach to trajectory planning of the wafer transport robotic arm. Using MATLAB simulation platform, it is shown that by introducing certain improvements to the snake optimization (SO) algorithm (a chaotic tent map for swarm initialization, using randomly perturbed dynamic learning factors, Bayesian optimization for parameterization and fine-tuning of the system selection process) and by introducing ISO into the Cartesian space of the robotic arm for trajectory planning, an S-shaped speed curve increased by 24.1% compared to the original plan and mean and variance ratings improved by 60.8% and 63.4%, respectively, compared to the SO algorithm. Original results, sufficiently significant and reliable conclusions, good quality figures and a detailed description of the data and methodology are presented; undoubtedly, the manuscript is of interest to a fairly wide range of readers. However, there are several significant comments on the organization of the manuscript in the Introduction and Statement of the Problem sections that seem necessary to take into account in order to make corrections and additions to improve the quality and readability of the manuscript, as detailed below.
First of all, the Introduction section should provide the most general formulation of the optimal trajectory planning problem, before discussing the state of the art of its achievements. Since different formulations of the trajectory planning problem are possible for different manipulators, it is desirable to change the structure of the Introduction section, formulating the general research problem at the beginning.
Secondly, there are comments on the formulation of the research problem. Specific information on the formulation of the research problem is given in Section 2.2. The Mathematical Model of Multiple Objective Function on page 5, lines 166-175, and immediately a number of questions arise that remain open when reading this section. Namely:
- What exactly is the input of the algorithm and what exactly is the output?
- Question about the input data
As far as can be understood from the manuscript, the input data for the trajectory planning procedure proposed by the authors are (generally speaking) a spatial trajectory on a plane and a set of constraints. How exactly is the spatial trajectory specified? Let us quote the text of the manuscript on lines 166-170: "Considering the production background and optimization effect, a multi-segment point-to-point trajectory in Cartesian space is adopted to achieve multi-objective optimization of time and impact. First, according to the expected trajectory of the robotic arm, it must be discretized into several interpolation points in Cartesian space." Should this be understood as meaning that the trajectory is specified by a sequence of vertices (each vertex is a pair of Cartesian coordinates of a point on a plane)? And if so, how are the interpolation points selected? In any case, it is necessary to describe precisely how the spatial trajectory is specified. In addition, the question arises: are there any restrictions on the deviations of the manipulator's trajectory, constructed using interpolation points, from the specified one? All this should be specified in this section.
- Question about the output of the algorithm.
It is not entirely clear what is supposed to be the main result of the algorithm proposed by the authors (the final goal)? Is it the representation of the trajectory by segments with the assignment of speed-curves parameters for each segment, or is it the construction of the trajectory in the joint space, or is it that for the optimal trajectory found in Cartesian coordinates, an exactly corresponding trajectory is constructed in the joint space (which is more likely)? There is a statement in lines 543-456 that seems not entirely clear: "The joint angle corresponding to the entire trajectory in the joint space can be calculated using the inverse kinematics equation of the robot. Finally, only the selected joint angle conforming to the motion trajectory is substituted into the forward kinematics of the robot to solve the cartesian coordinate value of the expected trajectory". Does this statement mean that some of the points of the joint space trajectory are excluded, and interpolation is performed for the remaining ones? If so, is it correct to talk about the exact correspondence of the trajectories on the plane and in the joint space? It would be desirable to clarify this statement.
- It is necessary to clarify the problem statement in two directions:
2.1 It is necessary to clarify the technical formulation of the problem (for example, manipulators with positional control require translation of the joint-space trajectory of the command sequence that the manipulator controller sends to the drives in the joints)
2.2 A clear mathematical formulation of the optimization problem is needed. The authors talk about constructing the objective function, but do not provide its domain of definition, that is, they do not specify how the segments on which the speed is optimized are selected.
Author Response
Comments 1: The Introduction section should provide the most general formulation of the optimal trajectory planning problem, before discussing the state of the art of its achievements. Since different formulations of the trajectory planning problem are possible for different manipulators, it is desirable to change the structure of the Introduction section, formulating the general research problem at the beginning.
Response 1: Thank you for pointing this out. I agree with this comment. Therefore, I have revised the content of the first paragraph to address the omission of the general research questions in the introduction. By adding the description of the optimal trajectory planning problem, the content of the following three parts is introduced, and the literature is replaced to support the manuscript discussion. The changes are on lines 32-40 of page 1.
Comments 2: There are comments on the formulation of the research problem. Specific information on the formulation of the research problem is given in Section 2.2. The Mathematical Model of Multiple Objective Function on page 5, lines 166-175, and immediately a number of questions arise that remain open when reading this section. Namely:
- What exactly is the input of the algorithm and what exactly is the output?
Question about the input data
As far as can be understood from the manuscript, the input data for the trajectory planning procedure proposed by the authors are (generally speaking) a spatial trajectory on a plane and a set of constraints. How exactly is the spatial trajectory specified? Let us quote the text of the manuscript on lines 166-170: "Considering the production background and optimization effect, a multi-segment point-to-point trajectory in Cartesian space is adopted to achieve multi-objective optimization of time and impact. First, according to the expected trajectory of the robotic arm, it must be discretized into several interpolation points in Cartesian space." Should this be understood as meaning that the trajectory is specified by a sequence of vertices (each vertex is a pair of Cartesian coordinates of a point on a plane)? And if so, how are the interpolation points selected? In any case, it is necessary to describe precisely how the spatial trajectory is specified. In addition, the question arises: are there any restrictions on the deviations of the manipulator's trajectory, constructed using interpolation points, from the specified one? All this should be specified in this section.
Question about the output of the algorithm.
It is not entirely clear what is supposed to be the main result of the algorithm proposed by the authors (the final goal)? Is it the representation of the trajectory by segments with the assignment of speed-curves parameters for each segment, or is it the construction of the trajectory in the joint space, or is it that for the optimal trajectory found in Cartesian coordinates, an exactly corresponding trajectory is constructed in the joint space (which is more likely)? There is a statement in lines 543-456 that seems not entirely clear: "The joint angle corresponding to the entire trajectory in the joint space can be calculated using the inverse kinematics equation of the robot. Finally, only the selected joint angle conforming to the motion trajectory is substituted into the forward kinematics of the robot to solve the cartesian coordinate value of the expected trajectory". Does this statement mean that some of the points of the joint space trajectory are excluded, and interpolation is performed for the remaining ones? If so, is it correct to talk about the exact correspondence of the trajectories on the plane and in the joint space? It would be desirable to clarify this statement.
-
It is necessary to clarify the problem statement in two directions:
2.1 It is necessary to clarify the technical formulation of the problem (for example, manipulators with positional control require translation of the joint-space trajectory of the command sequence that the manipulator controller sends to the drives in the joints)
2.2 A clear mathematical formulation of the optimization problem is needed. The authors talk about constructing the objective function, but do not provide its domain of definition, that is, they do not specify how the segments on which the speed is optimized are selected.
Response 2: Thank you for pointing this out. I agree with this comment.
Regarding the input data issue, the reviewer's understanding of spatial trajectories and kinematic constraints aligns with the manuscript content. The specification of spatial trajectories refers to the predefined motion trajectories constructed by the authors, which are elaborated in section 2.3.2. The lack of necessary descriptions in this section caused confusion for the reviewers. Therefore, I add explanatory statements in page 5, lines 172-176 to clarify the input data framework. Additionally, as correctly noted by the reviewer, the trajectory is defined by a series of discrete points. The selection of interpolation points is implemented through trajectory planning control code. Specifically, interpolation points are sampled every 0.002 seconds along the predefined motion trajectory. Given the extremely small time interval, the deviation between the actual executed trajectory and the interpolated trajectory can be considered negligible. To address the ambiguity in methodology description, detailed explanations of the interpolation point generation process have been incorporated in page 5, lines 181-186.
Regarding the issues with the algorithm output, the ultimate objective of this paper is to find the optimal values of acceleration and jerk within their respective constraint ranges by using an improved snake algorithm, with acceleration and jerk serving as independent variables. The algorithm outputs a determined acceleration and jerk. The ranges of acceleration and jerk values are also pre-defined by the authors. Meanwhile, the trajectory is fitted from several interpolation points spaced at 0.002-second intervals as mentioned above, so the precision can be guaranteed. Regarding the statement in the manuscript about excluding certain interpolation points, I have added relevant sentences in page 5, lines 195-196 to explain this. Since the inverse kinematics of the robot can generate two different solutions, this paper needs to exclude the solutions that do not conform to the expected trajectory. However, I did not explain this in detail in the manuscript but instead added this constraint in the code, so the lack of necessary description here may cause confusion for the reviewer.
Regarding the issues with technical descriptions, I have revised and added sentences in page 5, lines 180 - 196 to clarify the issues. In addition, only jerk needs a defined domain in the objective function, and the time interval is a fixed value of 0.002 seconds. Section 2.2 does not discuss a specific problem, so the manuscript does not specify the jerk domain. The domains of the objective function and the constraint conditions are both explained in section 3.2.
Reviewer 2 Report
Comments and Suggestions for Authors
The authors investigated a snake swarm algorithm to optimise trajectory optimisation of a robotic arm. In my opinion this is not a novel topic as it is an already existing method. So the following query has to be addressed.
1. The authors use a kinematic model of a robotic arm described at great length, if this is an already well known robotic arm, please just simplify the description.
2, The authors mention the use of chaotic tent mapping, dynamic learning factors and Bayesian optimisation to improve the ISO algorithm. However, then many literatures have proposed improvements such as multi-strategy chaotic systems. Please specify the difference between the improvement strategy of this paper and the existing ISO improvement methods, and add the literature comparison.
3、Bayesian optimisation is used for parameter fine-tuning, but often faces the risk of overfitting. Please detail the specific ways of integrating Bayesian optimisation in ISO (e.g., hyperparameter range, number of iterations) and explain how to avoid the negative impact of overfitting on practical applications.
4, The complexity introduced by the ISO algorithm (e.g., chaotic initialisation, dynamic learning factor) may increase the computational burden. Please analyse the feasibility of the algorithm in real-time control systems (e.g. single iteration time consuming)
5、Symmetric S-shaped curve is difficult to adapt to acceleration and deceleration asymmetric scenarios. Please clarify whether the improved S-shaped curve supports asymmetric parameter configurations and verify its robustness under abrupt trajectory changes.
Comments on the Quality of English Language
the english can be improved
Author Response
Comments 1: The authors use a kinematic model of a robotic arm described at great length, if this is an already well known robotic arm, please just simplify the description.
Response 1: Thank you for pointing this out. I agree with this comment. Therefore, I have made some cuts and changes to section 2.3.1. The formulas for the related displacement, acceleration and acceleration as well as the explanation are removed and only the important velocity curves are retained.
Comments 2: The authors mention the use of chaotic tent mapping, dynamic learning factors and Bayesian optimisation to improve the ISO algorithm. However, then many literatures have proposed improvements such as multi-strategy chaotic systems. Please specify the difference between the improvement strategy of this paper and the existing ISO improvement methods, and add the literature comparison.
Response 2: Thank you for pointing this out. I agree with this comment. Therefore, I add the differences between this improvement strategy and existing ISO improvement methods, and briefly compare the methods in section 2.4 of pages 9-13.
Comments 3: Bayesian optimisation is used for parameter fine-tuning, but often faces the risk of overfitting. Please detail the specific ways of integrating Bayesian optimisation in ISO (e.g., hyperparameter range, number of iterations) and explain how to avoid the negative impact of overfitting on practical applications.
Response 3: Thank you for pointing this out. I agree with this comment. Therefore, I elaborate on the specific methods for integrating Bayesian optimization in ISO and add two key experimental parameters. In addition, in order to effectively avoid the risk of overfitting, I introduce the K-fold cross-validation technique to enhance the estimation generalization ability of the model for unknown data. As a common method to prevent overfitting in Bayesian optimization, K-fold cross-validation can effectively improve the generalization performance of the model by dividing the data set into K subsets and conducting multiple training and validation. By introducing K-fold cross-validation, not only the robustness of the model is enhanced, but also the whole experimental process is more complete and reliable. The specific changes are in lines 498-520 on page 13 and lines 675-686 on page 21.
Comments 4: The complexity introduced by the ISO algorithm (e.g., chaotic initialisation, dynamic learning factor) may increase the computational burden. Please analyse the feasibility of the algorithm in real-time control systems (e.g. single iteration time consuming)
Response 4: Thank you for pointing this out. I agree with this comment. In response to the questions raised by the reviewers, I adjusted the relevant main parameters on the laptop using the MATLAB software and found that the feasibility of this method in real-time control systems is relatively low. Specifically, when the population size N = 10, the number of iterations T = 5, and the number of Bayesian optimization iterations is 10, the duration of a single iteration is 84 seconds, while the time for a single Bayesian optimization is controlled within 20 seconds; when the population size N = 20, the number of iterations T = 5, and the number of Bayesian optimization iterations is 10, the duration of a single iteration increases to 192 seconds, and the time for a single Bayesian optimization is within 70 seconds; when the population size N = 10, the number of iterations T = 10, and the number of Bayesian optimization iterations is 10, the duration of a single iteration is 120 seconds, and the time for a single Bayesian optimization is within 60 seconds. Due to the introduction of Bayesian optimization, the number of iterations of Bayesian optimization needs to be run in the first iteration process, and this mechanism's characteristic makes it difficult to achieve short-term real-time control. At the same time, the setting of the number of iterations for chaotic initialization, dynamic learning factors, and cosine annealing learning rate also affects the running time to varying degrees. Although these improved codes help enhance the algorithm's optimization ability and adaptability, they inevitably increase the consumption of computing resources. Therefore, this method can only be implemented by completing the calculation in the background and then retransmitting it to the controller. I am grateful for the reviewers' valuable suggestions. In the future, I will continue to strive to optimize the algorithm through algorithm improvement, hardware optimization, and parallel computing to reduce the running time of the algorithm to a certain extent and improve its feasibility in real-time control systems.
Comments 5: Symmetric S-shaped curve is difficult to adapt to acceleration and deceleration asymmetric scenarios. Please clarify whether the improved S-shaped curve supports asymmetric parameter configurations and verify its robustness under abrupt trajectory changes.
Response 5: Thank you for pointing this out. I agree with this comment. In the process of integrating the symmetric S-shaped speed curve with the ISO algorithm to adapt to asymmetric acceleration and deceleration scenarios, as indicated by the reviewers, the authors noted that the initial design did not address this specific case. However, the standalone S-shaped speed curve code supports asymmetric acceleration and deceleration as well as non-zero final velocity scenarios. This capability stems from its mechanism of dividing the trajectory into multiple segments, where the initial and final segments can independently utilize the S-shaped speed curve, while intermediate segments are connected at a constant velocity. Following the constructive feedback from the reviewers, the author has actively endeavored to integrate the S-shaped speed curve with the ISO algorithm to address the challenges posed by asymmetric acceleration and deceleration scenarios. Regrettably, due to time constraints, this integration has not yet been successfully implemented. Nonetheless, the author is profoundly appreciative of the reviewers' attention and valuable suggestions, and will continue to strive for improvements in future research and development efforts to better meet practical application needs. Once again, the author extends sincere apologies to the reviewers.
Round 2
Reviewer 1 Report
Comments and Suggestions for Authors
I am satisfied with the authors' responses to my comments and the additions made to the manuscript and believe that this manuscript can be recommended for publication in its present form.
Reviewer 2 Report
Comments and Suggestions for Authors
After a thorough review of the revised manuscript, the authors have fully addressed the initial review comments and made substantial improvements. The revised manuscript meets the publication standards of the journal, and I agree to accept it.